# Construction of a Glycaemia-Based Signature for Predicting Acute Kidney Injury in Ischaemic Stroke Patients after Endovascular Treatment

**DOI:** 10.3390/jcm11133865

**Published:** 2022-07-03

**Authors:** Chengfang Liu, Xiaohui Li, Zhaohan Xu, Yishan Wang, Teng Jiang, Meng Wang, Qiwen Deng, Junshan Zhou

**Affiliations:** 1Department of Neurology, Nanjing First Hospital, Nanjing Medical University, Nanjing 210006, China; liucf0212@163.com (C.L.); lixiaohui2288@163.com (X.L.); xxuzhaohan@sina.com (Z.X.); jiang_teng@njmu.edu.cn (T.J.); 15895826278@163.com (M.W.); 2Department of Laboratory Medicine, Nanjing First Hospital, Nanjing Medical University, Nanjing 210006, China; 13851501252@139.com

**Keywords:** acute kidney injury, endovascular treatment, glycaemia, nomogram, ischaemic stroke

## Abstract

Background: Hyperglycaemia is thought to be connected to worse functional outcomes after ischaemic stroke. However, the association between hyperglycaemia and acute kidney injury (AKI) after endovascular treatment (EVT) remains elusive. The purpose of this study was to investigate the influence of glycaemic on AKI after EVT. Methods: We retrospectively collected the clinical information of patients who underwent EVT from April 2015 to August 2021. Blood glucose after EVT was recorded as acute glycaemia. Chronic glucose levels were estimated by glycosylated haemoglobin (HbA_1c_) using the following formula: chronic glucose levels (mg/dL) = 28.7 × HbA_1c_ (%) − 46.7. AKI was defined as an increase in maximum serum creatinine to ≥1.5 baseline. We evaluated the association of AKI with blood glucose. A nomogram was established to predict the risk of AKI, and its diagnostic efficiency was determined by decision curve analysis. Results: We enrolled 717 acute ischaemic stroke patients who underwent EVT. Of them, 205 (28.6%) experienced AKI. Acute glycaemia (OR: 1.007, 95% CI: 1.003–1.011, *p* < 0.001), the acute/chronic glycaemic ratio (OR: 4.455, 95% CI: 2.237–8.871, *p* < 0.001) and the difference between acute and chronic glycaemia (Δ_A-C_) (OR: 1.008, 95% CI: 1.004–1.013, *p* < 0.001) were associated with the incidence of AKI. Additionally, age, atrial fibrillation, ASITN/SIR collateral grading, postoperative mTICI scale, and admission NIHSS were also significantly correlated with AKI. We then created a glycaemia-based nomogram, and its concordance index was 0.743. The net benefit of the nomogram was further confirmed by decision curve analysis. Conclusions: The glycaemia-based nomogram may be used to predict AKI in ischaemic stroke patients receiving EVT.

## 1. Introduction

Acute kidney injury (AKI) is an important complication of stroke, especially in patients receiving endovascular treatment (EVT). EVT is a vital therapeutic approach in patients with large vessel occlusion. Although it has been proven to be beneficial in reducing disability [1], nearly 50% of patients did not obtain favourable outcomes after successful recanalization [2]. The incidence of AKI after ischaemic stroke varies from 3.3% to 42.8% in patients undergoing EVT [3,4,5]. Previous studies have shown that AKI is associated with worse outcomes and increased mortality [3,4,5]. Diabetes mellitus (DM) is regarded as a risk factor for poor functional outcomes and AKI after EVT in ischaemic stroke patients [4]. Blood glucose is an important factor that can modify the treatment effect of thrombectomy on functional outcomes [6]. Admission hyperglycaemia is found to be present in 34.8% of stroke patients undergoing EVT [7] and is thought to be related to worse functional outcomes [8,9]. Blood glucose level changes dynamically and is considered an intensified postischemic inflammatory response [10]. It could be influenced by stroke severity [11] and may also be affected by EVT which is a kind of stress to the body. Persistent hyperglycaemia, which means hyperglycaemia at admission plus within 24 h after EVT, was found to be significantly associated with an increased risk of poor clinical outcomes [12]. Li et al. discovered that blood glucose levels within 24 h after EVT, rather than admission glucose, were independently connected with symptomatic intracranial haemorrhage [13]. It revealed that post-operative glycaemia played an important role in clinical prognosis. However, few studies discussed the effect of glycaemia after EVT. Additionally, random serum glucose is vulnerable to feeding and hypoglycaemic drugs [14]. Elevated fasting glucose concentration may be more reliable in assessing glucose metabolism [14] and was verified to be connected with adverse functional outcomes in EVT patients [15]. A higher stress hyperglycaemic ratio was found to be related to higher 90-day mortality, intensive care unit admission, and the need for mechanical ventilatory support during acute illness [16]. It was similar in acute ischaemic stroke patients. Stress hyperglycaemic ratio (SHR) calculated using the ratio of fasting blood glucose and glycosylated haemoglobin (HbA_1c_) could discriminate between chronic hyperglycaemia and acute glycaemia elevation and was found to be independently connected with three-month poor outcomes in thrombolytic or thrombectomy patients [17,18]. Limited data are available on the relevance of glycaemia after EVT to AKI. Therefore, we investigated the relationship between glycaemia and the incidence of AKI after EVT for ischaemic stroke.

## 2. Methods

We enrolled consecutive acute ischaemic stroke patients from our stroke register database who were admitted to Nanjing First Hospital, Nanjing Medical University and underwent EVT from April 2015 to August 2021. Patients lacking HbA_1c_ and serum creatinine levels were excluded. All participants’ baseline characteristics were collected, including age, sex, medical history, laboratory test results, National Institute of Health Stroke Scale (NIHSS) score, infarct circulation, stroke subtypes, collateral circulation, and details of the procedure, such as intravenous thrombolysis proportion, interval time, number of devices passed and recanalization status. Collateral circulation was assessed by the American Society of Interventional and Therapeutic Neuroradiology/Society of Interventional Radiology (ASITN/SIR). The scores are shown below: grade 0: no collaterals visible at the ischaemic site; grade 1: slow collaterals to the periphery of the ischaemic site with the persistence of some of the defect; grade 2: rapid collaterals to the periphery of the ischaemic site with the persistence of some of the defect and to only a portion of the ischaemic territory; grade 3: collaterals with slow but complete angiographic blood flow of the ischaemic bed by the late venous phase, and grade 4: complete and rapid collateral blood flow to the vascular bed in the entire ischaemic territory by retrograde perfusion [19]. Recanalization status was measured by the Modified Thrombolysis in Cerebral Infarction (mTICI) scale. Grade 0 means no perfusion. Grade 1 means penetration with minimal perfusion. Grade 2a means only partial filling of the entire vascular territory is visualized. Grade 2b means complete filling of all of the expected vascular territory is visualized, but the filling is slower than normal, and grade 3 means complete perfusion [19]. All procedures and medical care followed the current guidelines [20]. Informed consent was obtained from all participants for the EVT procedure, and the analysis was approved by the Ethics Committee of Nanjing First Hospital, Nanjing Medical University.

Blood glucose after EVT was recorded as acute glycaemia. HbA_1c_ was used to estimate the average chronic glucose level according to the validated formula: chronic glucose levels (mg/dL) = 28.7 × HbA_1c_ (%) − 46.7 [21]. All participants’ blood glucose and chronic glucose levels were utilized to calculate the acute/chronic glycaemic ratio and the absolute difference between acute and chronic glycaemia (Δ_A-C_). Acute glycaemia and HbA_1c_ were measured using the fasting blood sample after EVT. Serum creatinine was measured during hospitalization. The first measurement was conducted before EVT, and the second measurement was at the next morning after EVT and then assayed according to the change of the disease. The minimum creatinine level during hospitalization was regarded as baseline creatinine. AKI was defined as an increase in maximum serum creatinine to ≥1.5 baseline over seven days according to the Kidney Disease Improving Global Guidelines (KDIGO) Clinical Practice Guidelines [22].

## 3. Statistical Analysis

Continuous data are presented as the mean ± standard deviation or median with quartiles and were compared with Student’s t test or the Mann–Whitney U test if necessary. Categorical data are presented as numbers with percentages and were analysed by the chi-squared test. Variables with *p* < 0.1 in univariate analyses were selected for multivariable logistic regression analysis to distinguish the risk factors associated with AKI. Receiver operating characteristic (ROC) curves and the areas under the receiver operating characteristic curves were calculated to measure the ability of variables to predict AKI. All the significant variables were then converted to a nomogram to predict the risk of AKI. The forecast performance was examined by the concordance index (c-index) and calibration plot, which clarified the association between the actual and predicted probability. Diagnostic efficiency was evaluated by decision curve analysis, which compared the net benefits of each prediction model. A two-tailed *p* < 0.05 indicated statistical significance. Statistical analyses were performed using the Statistical Package for the Social Sciences (SPSS) version 20.0 (SPSS Inc., Chicago, IL, USA) and R software version 3.5.2 (Institute for Statistics and Mathematics, Vienna, Austria).

## 4. Results

From April 2015 to August 2021, a total of 788 patients received EVT in our centre. Among them, 71 lacked a(n) HbA_1c_ or serum creatinine level and were excluded, and 717 were enrolled in the final analysis. The average age of the cohort was 70.2 ± 11.9 years, and 64.3% of patients were male. The mean acute glucose level was 128 ± 45 mg/dL, and the estimated chronic glucose level was 135 ± 40 mg/dL (Table 1). Of the 717 patients with ischaemic stroke, 205 (28.6%) experienced AKI (Figure 1). Table 1 shows the baseline characteristics of the patients with AKI and without AKI. Significant differences were detected between the two groups with respect to age (74.5 ± 11.0 years vs. 68.6 ± 12.0 years, *p* < 0.001), history of atrial fibrillation (61.0% vs. 38.7%, *p* < 0.001), acute glycaemia (143 ± 50 mg/dL vs. 123 ± 42 mg/dL, *p* < 0.001), serum creatinine (83.3 ± 39.7 μmol/L vs. 75.2 ± 29.5 μmol/L, *p* = 0.003), baseline NIHSS [17 (13–21) vs. 13 (10–18), *p* < 0.001], ASITN/SIR [1 (1–2) vs. 2 (1–2), *p* < 0.001], prior intravenous thrombolysis (36.1% vs. 44.5%, *p* =0.039), number of devices passed [2 (1–3) vs. 2 (1–3), *p* = 0.002], and mTICI score (81.0% vs. 89.8%, 19.0% vs. 10.2%, *p* = 0.002).

Table 2 shows that AKI risk was significantly higher when the acute glycaemia (OR: 1.007, 95% CI: 1.003–1.011, *p* < 0.001), acute/chronic glycaemic ratio (OR: 4.455, 95% CI: 2.237–8.871, *p* < 0.001), and Δ_A-C_ (OR: 1.008, 95% CI: 1.004–1.013, *p* < 0.001) were considered compared with that associated with chronic glycaemia (OR: 1.001, 95% CI: 0.997–1.005, *p* = 0.715). Prespecified subgroup analyses are shown in Appendix A, and we found a similar conclusion in patients with or without DM. ROC analyses showed that acute glycaemia, the acute/chronic glycaemic ratio, and Δ_A-C_ also exhibited greater accuracy than average chronic glucose levels in terms of the capacity to predict AKI (Appendix A).

Age, atrial fibrillation, ASITN/SIR collateral grading, postoperative mTICI scale, admission NIHSS, acute glycaemia, the acute/chronic glycaemic ratio, and Δ_A-C_ were identified as independent risk factors in logistic regression analyses (Table 2). We created a nomogram model to precisely predict AKI risk using these factors (Figure 2). In the nomogram, each predictor had a parallel score within the range of 0 to 100 for its contribution to AKI, which could be used to forecast the risk of AKI. The nomogram had a concordance index of 0.743, whereas the value was 0.722 when the latter three factors were not considered. The calibration curve showed good consistency between the predicted and actual risk of AKI (Figure 3). The net benefits between the combined factors versus without glycaemia in predicting the probability of AKI were assessed by decision curve analysis (Figure 4). In this analysis, the former method provided a higher net benefit in a wide range, suggesting that it was a superior forecasting method.

## 5. Discussion

Our study showed the ability of the acute glucose level, acute/chronic glycaemic ratio, and Δ_A-C_ to predict AKI after EVT for ischaemic stroke. Furthermore, we constructed a nomogram prediction model based on significant variables (age, atrial fibrillation, ASITN/SIR collateral grading, postoperative mTICI scale, admission NIHSS, acute glycaemia, acute/chronic glycaemic ratio, and Δ_A-C_), which showed good accuracy in forecasting the probability of AKI, with a concordance index of 0.743. It was well calibrated, and the decision curve analysis suggested that it had clinical utility.

AKI is an important complication after acute stroke. Several studies have proven that AKI increases the risk of death and poor functional outcomes in the short term [3,4,23]. It is also an independent prognostic factor for long-term (10-year) mortality and new composite cardiovascular events [24]. However, the incidence of AKI in patients receiving EVT differed between studies, ranging from 3.3% to 42.8% [3,4,5]. Some researchers have reported that AKI is uncommon in patients receiving EVT [3], whereas others found it is a common complication after acute stroke [5]. A systematic review showed that the rates of AKI after stroke varied from 0.82% to 26.68%, and the pooled incidence was 9.61% [23]. It found that studies using creatinine definitions reported that the AKI incidence was 19.5%, whereas others using the International Classification of Diseases-9th or 10th Edition coding definitions reported that the rate was 4.63%. The marked difference may be partly attributed to the methodology used to identify AKI. The latter method is thought to underestimate the AKI rate due to its low sensitivity [25]. In the present study, 205 (28.6%) patients experienced AKI after EVT. The high incidence of AKI may have been due to the older age in the sample (70.2 ± 11.9 vs. 63.9 ± 15.8) [3], which was similar to the findings from Simon et al. [5]. The final AKI sample included patients who experienced AKI at admission and in the hospital according to the definition of AKI from Simon et al.

DM is an independent risk factor for stroke, which can cause the hardening of arteries and the degeneration of artery compliance [26]. A review pointed out that patients with a history of DM who received EVT had significantly lower odds of functional independence [27]. Jun Young Chang et al. found that the incidence of early neurological deterioration and symptomatic haemorrhage, as well as the proportion of patients with a modified Rankin scale score > 1, were significantly higher in diabetic patients who underwent EVT with higher pre-stroke glucose levels, which was reflected by HbA_1c_ [28]. DM is also related to AKI [4,29]. Whereas the specific pathophysiologic mechanisms of AKI remain unclear, it cannot be denied that DM leads to renal microangiopathy. While some other researchers did not find DM to be significantly associated with AKI [30,31]. In our study, the incidence of DM was similar in patients with AKI (33.2%) and without AKI (31.6%). Multivariable logistic regression analysis showed no significance of chronic glycaemia in predicting AKI, whether they were diabetic or not. This may have been partly due to the similar HbA_1c_ in patients with ischaemic stroke. However, hypoglycaemic therapy could reduce HbA1c levels [32]. In the present study, we did not find the percentage of taking antidiabetic drugs was different between patients with or without AKI.

Hyperglycaemia is common in acute ischaemic stroke patients. A post hoc analysis showed that hyperglycaemia was independently associated with worse functional outcomes at three months in patients treated with mechanical thrombectomy, especially those with incomplete reperfusion [9]. Although Osei et al. found no evidence for an interaction between increased blood glucose and functional outcomes in patients undergoing EVT, the occurrence of adverse functional outcomes, symptomatic intracerebral haemorrhage, and mortality were higher in hyperglycaemic patients [33]. Rinkel et al. found that hyperglycaemia on admission was associated with a shift towards worse functional outcomes compared with nonhyperglycaemic patients, and there was a J-shaped association between admission glucose levels and poor outcomes [8]. They all defined hyperglycaemia as admission serum glucose > 140 mg/dL and confirmed the inverse interaction between hyperglycaemia and functional outcomes, but no one explored whether the association still existed in the presence of AKI. Hyperglycaemia is the response of stress caused by cerebral ischaemia and is related to stroke severity [11]. Thus, the blood sugar level changes dynamically and could be impacted by several factors like EVT. We collected fasting blood glucose the next morning after EVT to calculate the ratio and absolute difference between acute and chronic glycaemia. We found that patients with AKI always had higher serum blood glucose levels (143 ± 50 mg/dL), which met the above hyperglycaemia criteria. After adjustment, acute glycaemia remained associated with AKI. When combined with chronic glycaemia, it also showed good clinical value in predicting AKI. The acute/chronic glycaemia ratio considers acute glycaemic fluctuation and chronic glycaemia status and represents relative changes in blood glucose. Previous studies have revealed that high SHR was associated with poor outcomes in thrombolytic or thrombectomy patients [17,18]. The absolute difference between acute and chronic glycaemia was recorded as Δ_A-C_ which could also be termed as glycaemic gap. Yang et al. found that both SHR and glycaemic gap were useful in evaluating the ischaemic stroke severity and prognosis [34]. However, Roberts et al. discovered that SHR instead of glycaemic gap was associated with outcomes in ischaemic stroke patients [35]. We found that both the acute/chronic glycaemic ratio and Δ_A-C_ were significant predictors of AKI, and the risk of AKI was four times higher for every 0.1 increasement in the acute/chronic glycaemic ratio. It indicated that a relative blood glucose increase had a more detrimental effect on renal function. The mechanism between hyperglycaemia and AKI in patients receiving EVT is unidentified. However, prior researches have demonstrated that increased oxidative stress, blunting of free radical scavengers, decreased levels of nitric oxide, and endothelial dysfunction are activated in the setting of elevated blood glucose in patients with acute coronary syndrome [36,37]. It could also explain how AKI risk is increased with elevated blood glucose to some degree while further research and exploration are needed.

Although DM is a risk factor for AKI after EVT in ischaemic stroke patients [4], few studies have examined the association between blood glucose and AKI occurrence. Stolker et al. discovered that elevated preprocedural glucose is related to a greater risk of contrast-induced AKI in acute myocardial infarction patients without known diabetes after coronary angiography [36]. This finding showed us that the relevance might also exist in stroke patients. In the present study, we found a significant increase in AKI occurrence tailed with acute glycaemia, acute/chronic glycaemic ratio, and Δ_A-C_ increase in patients treated with EVT. We showed similar prediction accuracy for these parameters in the ROC curve analysis. Wang et al. recruited 4527 patients hospitalized for atrial fibrillation and found that the incidence of in-hospital AKI in patients with atrial fibrillation was 2.6 times higher than that in patients with sinus rhythm [38]. Atrial fibrillation is considered to be related to adverse renal outcomes and causes AKI by several mechanisms such as hemodynamic perturbations and renal ischaemia from an embolic event [39,40]. We also found that there were 55% higher odds of AKI for patients with atrial fibrillation. Age, atrial fibrillation, ASITN/SIR collateral grading, postoperative mTICI scale, and admission NIHSS were also independent risk factors for AKI. Based on this discovery, we proposed a nomogram that could provide neurologists with the precise probability of AKI after intravascular treatment for large vessel occlusion patients. It offered potential clinical implications by assessing acute and chronic glucose levels in EVT patients to discern those who are at high risk of AKI and need careful monitoring of renal function after EVT. Moreover, it can supply strategies for blood glucose management to avoid AKI incidence and improve the prognosis of stroke patients. Current guidelines for glycaemia regulation of acute stroke patients are from the American Heart Association/American Stroke Association (AHA/ASA), which recommends maintaining the blood glucose level in the range of 140 to 180mg/dL [20]. Van den Berghe et al. found that keeping blood glucose < 110 mg/dL was in favour of preventing kidney injury in mixed medical/surgical intensive care units [41]. Schetz et al. also showed that the renoprotective effect was most pronounced in patients who achieved a mean morning blood glucose level < 110 mg/dL [42]. Our study indicated that it may be beneficial to avoid AKI by maintaining fasting blood glucose in a range of 80 to 123 mg/dL, but the specific range needs more researches to verify in the future.

To our knowledge, this is the first study to explore the relationship between acute and chronic blood glucose and AKI in patients receiving EVT. However, our study had several limitations. First, it was a single-centre retrospective study that might have been influenced by selection bias. Second, we determined that AKI occurred if the maximum serum creatinine level increased to ≥1.5 baseline over seven days according to KDIGO clinical practice guidelines [22]. Although it has high sensitivity, its specificity is low. Plasma creatinine levels might fluctuate due to different situations, such as medications, hypotension, or fluid shifts [43]. Dehydration or diuresis may also induce the elevation of serum glucose while we lacked relevant information. Additionally, we did not know the patients’ premorbid kidney function and took the minimum creatinine level during hospitalization as the baseline. Thus, we might have overestimated the incidence of AKI. Third, we did not investigate the effects of hypoglycaemic therapy and in-hospital blood glucose fluctuations on AKI. Fourth, we could not affirm that there was a cause-effect association between serum glucose and AKI. Nevertheless, this is an innovative finding revealing the hazardous factors of AKI in patients receiving intravascular treatment. Future research is needed to find more specific markers of AKI and verify the appropriate prediction model for AKI.

## 6. Conclusions

Among acute ischaemic stroke patients treated with EVT, the combined evaluation of acute and chronic glycaemia can predict AKI. We also developed a model that has forecasting value for the occurrence of AKI. The current findings can provide evidence for the management of blood glucose in intravascular treatment patients.

## Figures and Tables

**Figure 1 jcm-11-03865-f001:**
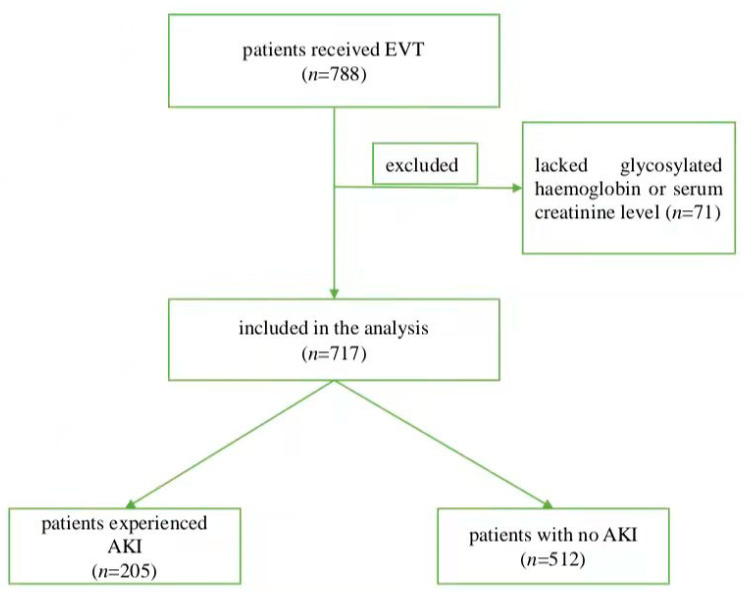
Study flow diagram. Abbreviations: EVT, endovascular treatment; AKI, acute kidney injury.

**Figure 2 jcm-11-03865-f002:**
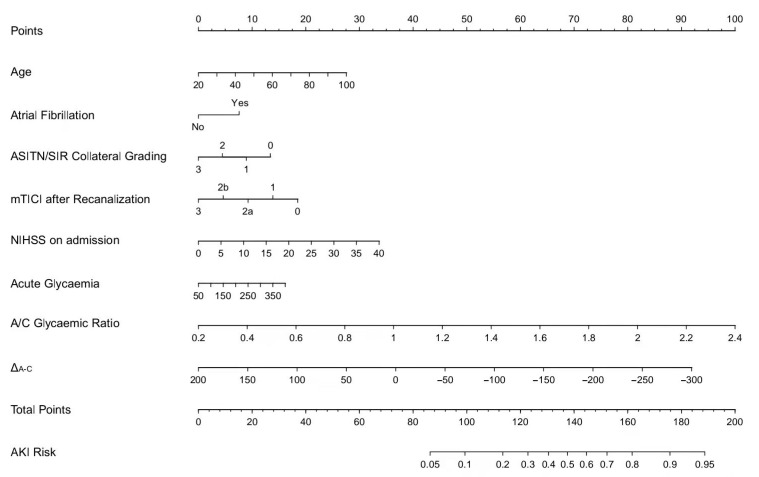
Nomogram predicting AKI in acute ischaemic stroke patients receiving EVT.

**Figure 3 jcm-11-03865-f003:**
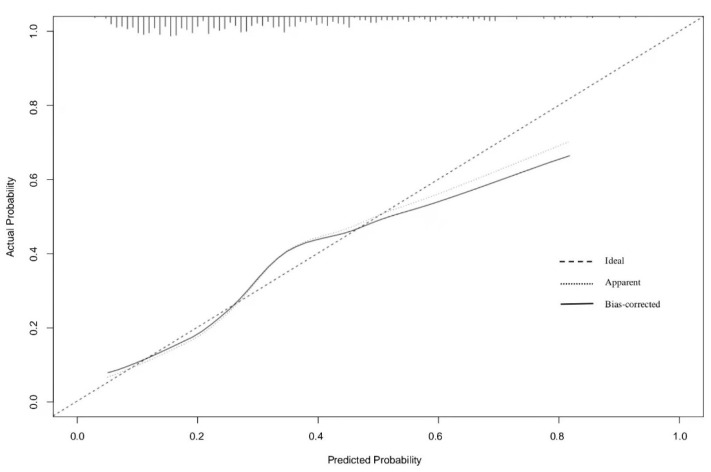
Calibration curve of the nomogram for predicting AKI.

**Figure 4 jcm-11-03865-f004:**
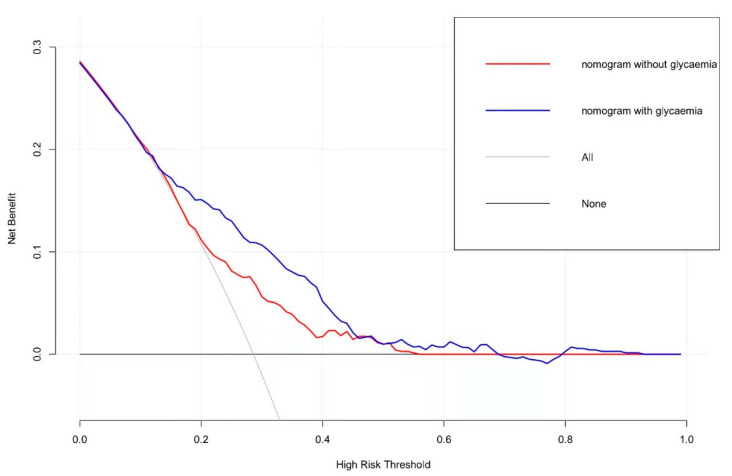
Decision curve analysis for the nomogram.

**Table 1 jcm-11-03865-t001:** Baseline characteristics of patients with and without AKI.

	Total (717)	AKI (*n* = 205)	No AKI (*n* = 512)	*p*
Age (years), mean ± SD	70.2 ± 11.9	74.5 ± 11.0	68.6 ± 12.0	<0.001
Sex, male, *n* (%)	461 (64.3%)	124 (60.5%)	337 (65.8%)	0.178
Medical history, *n* (%)				
Hypertension	539 (75.2%)	163 (79.5%)	376 (73.4%)	0.089
Diabetes	230 (32.1%)	68 (33.2%)	162 (31.6%)	0.692
Atrial fibrillation	323 (45.0%)	125 (61.0%)	198 (38.7%)	<0.001
Prior stroke	149 (20.8%)	37 (18.1%)	112 (21.9%)	0.261
Laboratory examination, mean ± SD				
Acute glycaemia, mg/dL	128 ± 45	143 ± 50	123 ± 42	<0.001
HbA1c, %	6.3 ± 1.4	6.3 ± 1.3	6.3 ± 1.3	0.716
Average chronic glycaemia, mg/dL	135 ± 40	136 ± 40	135 ± 40	0.716
A/C glycaemic ratio	0.97 ± 0.28	1.08 ± 0.31	0.93 ± 0.25	<0.001
Δ_A-C_, mg/dL	−6 ± 43	7 ± 47	−12 ± 40	<0.001
Baseline serum creatinine, μmol/L	77.5 ± 32.9	83.3 ± 39.7	75.2 ± 29.5	0.003
eGFR, mL/min/1.73 m^2^	99 ± 34	91 ± 32	101 ± 35	<0.001
Total cholesterol, mg/dL	76 ± 21	76 ± 23	77 ± 21	0.856
Triglycerides, mg/dL	23 ± 16	23 ± 20	23 ± 14	0.839
HDL, mg/dL	20 ± 6	21 ± 6	20 ± 6	0.855
LDL, mg/dL	46 ± 17	45 ± 18	46 ± 17	0.856
antidiabetic drugs, *n* (%)	100 (13.9%)	24 (11.7%)	76 (14.8%)	0.273
Baseline NIHSS score, median (IQR)	14 (11–19)	17 (13–21)	13 (10–18)	<0.001
Infarct circulation, *n* (%)				0.514
Anterior	608 (84.8%)	171 (83.4%)	437 (85.4%)	
Posterior	109 (15.2%)	34 (16.6%)	75 (10.5%)	
Stroke subtypes, *n* (%)				<0.001
LAA	327 (45.6%)	66 (32.2%)	261 (51.0%)	
CE	336 (42.9%)	130 (63.4%)	206 (40.2%)	
SOE	21 (2.9%)	2 (1.0%)	19 (3.7%)	
SUE	33 (4.6%)	7 (3.4%)	26 (5.1%)	
ASITN/SIR, median (IQR)	2 (1–2)	1 (1–2)	2 (1–2)	<0.001
Interval time, min, median (IQR)				
Onset to door	171 (85–300)	160 (70–297)	175 (90–300)	0.353
Door to groin puncture	105 (80–140)	110 (80–143)	105 (79–137)	0.692
Door to first recanalization	181 (148–225)	190 (155–242)	180 (145–220)	0.163
Intravenous thrombolysis, *n* (%)	302 (42.1%)	74 (36.1%)	228 (44.5%)	0.039
Number of devices passed, median (IQR)	2 (1–3)	2 (1–3)	2 (1–3)	0.002
mTICI score, *n* (%)				0.002
2b-3	626 (87.3%)	166 (81.0%)	460 (89.8%)	
0–2a	91 (12.7%)	39 (19.0%)	52 (10.2%)	

Abbreviations: AKI, acute kidney injury; SD, standard deviation; HbA_1c_, glycosylated haemoglobin; A/C, acute/chronic; Δ_A-C_, the difference between acute and chronic glycaemia; eGFR, estimated glomerular filtration rate; HDL, high density lipoprotein; LDL, low-density lipoprotein; NIHSS, National Institutes of Health Stroke Scale; IQR, interquartile range; LAA, large artery atherosclerosis; CE, cardiac embolism; SOE, stroke of other determined aetiology; SUE, stroke of undetermined aetiology; ASITN/SIR, American Society of Interventional, and Therapeutic Neuroradiology/Society of Interventional Radiology; mTICI, Modified Thrombolysis in Cerebral Infarction.

**Table 2 jcm-11-03865-t002:** Unadjusted and adjusted ORs of risk factors for AKI.

	Crude OR (95% CI)	*p*	Adjusted OR (95% CI)	*p*
Acute glycaemia	1.009 (1.006–1.013)	<0.001	1.007 (1.003–1.011)	<0.001
Chronic glycaemia	1.001 (0.997–1.005)	0.715		
A/C glycaemic ratio	7.333 (3.957–13.588)	<0.001	4.455 (2.237–8.871)	<0.001
Δ_A-C_	1.012 (1.007–1.016)	<0.001	1.008 (1.004–1.013)	<0.001
Age	1.044 (1.028–1.061)	<0.001	1.024 (1.006–1.043)	0.010
Hypertension	1. 404 (0.949–2.077)	0.090		
Atrial fibrillation	2.478 (1.777–3.454)	<0.001	1.555 (1.027–2.354)	0.037
Baseline NIHSS score	1.072 (1.049–1.096)	<0.001	1.049 (1.025–1.074)	<0.001
Stroke subtypes	1.091 (0.927–1.286)	0.295		
ASITN/SIR	0.553 (0.432–0.707)	<0.001	0.695 (0.531–0.911)	0.008
Intravenous thrombolysis	0.704 (0.504–0.983)	0.039	0.725 (0.503–1.043)	0.083
Number of devices passed	1.262 (1.115–1.428)	<0.001	1.107 (0.960–1.276)	0.161
mTICI score	0.481 (0.306–0.756)	0.001	0.563 (0.336–0.941)	0.029

Adjusted for age, hypertension, atrial fibrillation, baseline NIHSS score, stroke subtypes, ASITN/SIR, intravenous thrombolysis, number of devices passed, and mTICI score; Abbreviations: AKI, acute kidney injury; OR, odds ratio; A/C, acute/chronic; Δ_A-C_, the difference between acute and chronic glycaemia; NIHSS, National Institutes of Health Stroke Scale; ASITN/SIR, American Society of Interventional, and Therapeutic Neuroradiology/Society of Interventional Radiology; mTICI, Modified Thrombolysis in Cerebral Infarction.

## Data Availability

The datasets generated during and/or analysed during the current study are available from the corresponding author on reasonable request.

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
