# Peer review of "Construction of a Glycaemia-Based Signature for Predicting Acute Kidney Injury in Ischaemic Stroke Patients after Endovascular Treatment"

_jcm, 2022, doi:10.3390/jcm11133865_

Round 1

Reviewer 1 Report

The present study represents a well conducted analysis on the impact of hyperglycaemia for the development of AIK in patients with AIS underwent EVT. Often neglected among neurologists, especially in the acute setting, AIK has already been showed to significantly worsen medium-long term prognosis of AIS patients. The present work further highlights the association between hyperglycaemia and the risk of AIK development in a large cohort of AIS patients underwent EVT. 

Major Revision:

1)    Although the association between elevated blood glucose levels and worse functional outcome in AIS patients undergoing both IVT and EVT is well known, discrepancies regarding the most appropriate way to determine what “acute hyperglycaemia” is, still exist. For instance, in DOI: 10.3390/jcm9061932 the authors found that persistent elevated blood glucose levels after EVT rather than isolated pre-procedural glucose determination were associated with a poorer clinical outcome, suggesting that persistent hyperglicaemia might be more detrimental. In contrast, other works (for instance: DOI 10.1016/j.diabet.2016.07.036 and DOI: 10.3389/fneur.2021.725002) employed an index able to evaluate the level of “stress hyperglycaemia” showing that the higher it is, the worse the clinical outcome. Please try to expand the introduction section addressing this topics and clarifying the reason why you have decided to use your method among the others.

2)    It does not appear entirely clear to me the usefulness of calculating both the A/C glycaemic ratio and the delta A-C. Please try to better address the difference between these two measures.   

3)    According to Tab. 1, roughly the 30% of the population was affected by DM. Although the distribution of DM patients between the two groups was not significantly different, the determination of how many DM patients were on antidiabetic drugs would be of paramount importance considering that antidiabetic therapy may lower HbA1c value thus altering the determination of chronic glucose level and consequently having an impact on all the measures used. Please report this information if available and address this issue in the discussion section.    

4)    The prevalence of AF is greater within the AKI group and the presence of AF effectively predict the outcome in the regression analysis. Please provide a possible explanation regarding this topic in the discussion section.

Minor Revisions:

1)     In the methods section, please provide a brief description of both ASITN/SIR and mTICI scoring grades.

2)     In Tab. 1, numbers of device passed is not different between the two groups but the p-value appears to be significant. Please correct.

In Tab. 2 you reported both the crude and adjust OR for all the predictors taken into account. It does not look clear which covariates you added to the regression model for each predictor. Please address this issue.

Author Response

Thank you for your letter and for the reviewer’s comments concerning our manuscript. Those comments are all valuable and very helpful for revising and improving our paper. We have studied comments carefully and have made modification which we hope meet with approval.

Major Revision:

1. Although the association between elevated blood glucose levels and worse functional outcome in AIS patients undergoing both IVT and EVT is well known, discrepancies regarding the most appropriate way to determine what “acute hyperglycaemia” is, still exist. For instance, in DOI: 10.3390/jcm9061932 the authors found that persistent elevated blood glucose levels after EVT rather than isolated pre-procedural glucose determination were associated with a poorer clinical outcome, suggesting that persistent hyperglycaemia might be more detrimental. In contrast, other works (for instance: DOI 10.1016/j.diabet.2016.07.036 and DOI: 10.3389/fneur.2021.725002) employed an index able to evaluate the level of “stress hyperglycaemia” showing that the higher it is, the worse the clinical outcome. Please try to expand the introduction section addressing this topic and clarifying the reason why you have decided to use your method among the others.

Response: Thanks a lot for your suggestion. Previous studies have proved that admission hyperglycaemia is associated with poor outcome. But other studies pointed out that hyperglycaemia after EVT or persistent hyperglycaemia might be more detrimental. However, limited data are available on the relevancy of glycaemia after EVT to AKI. We took fasting blood glucose after EVT as acute glycemia for it partly reflected the stress response of the body to EVT and maybe more reliable in assessing glucose metabolism compared with random serum glucose. Higher stress hyperglycaemic ratio was found to be related to higher 90-day mortality, ICU admission and need for mechanical ventilatory support during acute illness. It was similar in stroke patients receiving EVT. The ratio calculated with fasting glucose and glycosylated haemoglobin (HbA1c) could discriminate between chronic hyperglycaemia and acute glycemia elevation. It was found to be associated with poor prognosis. However, they often investigated the relationship between an isolated blood glucose index and prognosis. Thus, we combined fasting blood glucose with HbA1c to evaluate their effects on AKI through calculating the ratio and difference of the aforesaid blood glucose indices. We have made some modification in the introduction part according to the papers you provided.

2. It does not appear entirely clear to me the usefulness of calculating both the A/C glycaemic ratio and the delta A-C. Please try to better address the difference between these two measures.   

Response: Thanks a lot for your suggestion. ΔA-C reflects absolute acute glycaemic changes which did not take chronic glucometabolic into account. Acute/chronic glycaemia ratio considers acute glycaemic fluctuation and chronic glycaemia status and represents relative changes of blood glucose. SHR was regarded to be associated with poor outcome while the significance of ΔA-C was controversial. In our study, we found that they both connected with AKI and AKI risk was four times higher for every 0.1 increasement in A/C glycaemic ratio. It indicated that relative blood glucose increase had a more detrimental effect on renal function. We have made some modification in the discussion part.

3. According to Tab. 1, roughly the 30% of the population was affected by DM. Although the distribution of DM patients between the two groups was not significantly different, the determination of how many DM patients were on antidiabetic drugs would be of paramount importance considering that antidiabetic therapy may lower HbA1c value thus altering the determination of chronic glucose level and consequently having an impact on all the measures used. Please report this information if available and address this issue in the discussion section.    

Response: Thanks a lot for your suggestion. We have added the percentage of taking antidiabetic drugs and there was no difference between patients with or without AKI. We also made some modification in the discussion part.

4. The prevalence of AF is greater within the AKI group and the presence of AF effectively predict the outcome in the regression analysis. Please provide a possible explanation regarding this topic in the discussion section.

Response: Thanks a lot for your suggestion. Previous studies pointed out that AF could increase the risk of AKI. We also found that the risk of AKI was 55% higher for patients with AF. Possible mechanisms may be hemodynamic perturbations and renal ischemia from an embolic event. We have made some modification in the discussion part.

 Minor Revisions:

1. In the methods section, please provide a brief description of both ASITN/SIR and mTICI scoring grades.

Response: Thanks for the suggestion of reviewer. We have made some modification in the method part.

2. In Tab. 1, numbers of device passed is not different between the two groups but the p-value appears to be significant. Please correct.

Response: Thanks for the suggestion of reviewer. We recalculated the numbers of device passed between the two groups and we found the same result as shown following. We also could provide the raw data in another Excel profile as a supporting file.

AKI

5

10

25

50

75

90

95

numbers of device passed

NO

0

1

1

2

3

4

4

YES

1

1

1

2

3

4

5

3. In Tab. 2 you reported both the crude and adjust OR for all the predictors taken into account. It does not look clear which covariates you added to the regression model for each predictor. Please address this issue.

Response: Thanks for the suggestion of reviewer. We have pointed out that ‘Variables with p<0.1 in univariate analyses were selected for multivariable logistic regression analysis to distinguish the risk factors associated with AKI’ in the method part. Also, the annotations below the Table 2 displayed the qualified variables.

Reviewer 2 Report

The authors retrospectively reviewed 717 cases of acute ischemic stroke who underwent mechanical thrombectomy and concluded that a "glycemia based normogram" is useful in predicting AKI in these patients. I have a few concerns and questions regarding this study. What's the main clinical application of this normogram and how would this make a difference in clinical practice? What is the mechanistic link between acute hyperglycemia and AKI? As the authors mentioned, SHINE trial did not find any outcome benefit in tight blood glucose control in patients with acute ischemic stroke. This would raise the possibility of other mechanisms to explain the findings of the current study. Could mild elevation of serum glucose be simply caused by dehydration or diuresis? The authors showed that atrial fibrillation was more common in the AKI group. I wonder if use of diuretics contributed to AKI in these patients. 

Author Response

The authors retrospectively reviewed 717 cases of acute ischemic stroke who underwent mechanical thrombectomy and concluded that a "glycemia based normogram" is useful in predicting AKI in these patients. I have a few concerns and questions regarding this study. What's the main clinical application of this normogram and how would this make a difference in clinical practice? What is the mechanistic link between acute hyperglycemia and AKI? As the authors mentioned, SHINE trial did not find any outcome benefit in tight blood glucose control in patients with acute ischemic stroke. This would raise the possibility of other mechanisms to explain the findings of the current study. Could mild elevation of serum glucose be simply caused by dehydration or diuresis? The authors showed that atrial fibrillation was more common in the AKI group. I wonder if use of diuretics contributed to AKI in these patients. 

Response: Thanks a lot for your suggestion.

1. We established a nomogram with a good degree of differentiation and accuracy. It could provide an individual assessment of AKI risk by gathering personal biological and clinical data. It could distinguish patients who need careful monitoring of renal function after EVT. However, it still needs external validation.

2. The mechanism between hyperglycemia and AKI in patients receiving EVT is unidentified. It may be attributed to increased oxidative stress, blunting of free radical scavengers, decreased levels of nitric oxide, and endothelial dysfunction. We have made some modification in the discussion part.

3. Hypoglycaemic therapy was found to be in favor of preventing kidney injury in intensive care units. The benefit may also exist for stroke patients but we need more researches to verify in the future.

4. Plasma creatinine levels might fluctuate due to different situations, such as medications, hypotension or fluid shifts. Dehydration or diuresis may also induce elevation of serum glucose while we lacked relevant information. It is also a limitation of our study and we have made some modification in the limitation section of discussion.

Round 2

Reviewer 1 Report

I am pleased with the corrections made by the authors which further improve the quality of the manuscript. 

In my opinion, the present work can be considered for publication.